# Synthetic fascia for stiff and tough 4D printed multifunctional structures that detect and tolerate damage

Javier M. Morales Ferrer[1], Chloe Kekedjian [2], Nicole Bacca[1] &
J. William Boley [1,2] ✉

Creating shape-morphing structures with time-responsive materials is a key goal of 4D printing, but combining high stiffness ($E$) and toughness ($K$) in a single material remains difficult. Soft materials stretch but lack strength, while stiff materials resist deformation but are brittle. Nature overcomes this trade-off in skeletal muscle by surrounding strong fibers with soft, tough tissue for protection and support. Inspired by this strategy, we develop a multi-material printing method that combines stiff synthetic muscle with a soft, stretchable adhesive to form a composite structure. Here, we show that this design greatly improves toughness without sacrificing stiffness, enables controlled actuation, and maintains function after multiple fractures. We demonstrate these properties in a damage-tolerant actuator, a lifting robot with record performance, and a lattice that detects and withstands extreme loads while remaining operational.

4D printing refers to an advanced form of additive manufacturing in which a 3D printed object is designed to undergo controlled, time-dependent changes in shape, properties, or function in response to external stimuli such as temperature, light, humidity, magnetic fields. This stimuli-responsive behavior distinguishes 4D printing from conventional static 3D structures, enabling dynamic and functional transformations. While many 4D printed systems are designed for reversible shape changes, certain applications exploit irreversible transformations to achieve stable, deployed configurations, expanding the functional scope of 4D printed architectures[1,2]. Previous studies have demonstrated that adopting this manufacturing approach leads to a substantial reduction in both printing time and materials consumption[1]. Furthermore, it facilitates the achievement of greater levels of geometric complexity and responsiveness in the resulting structures[3–5], a capability that remains beyond the reach of conventional manufacturing methods[6]. These morphing structured systems could potentially be used in a range of applications, from deployable systems[3,7,8] and dynamic optics[9,10], to robotics[5,11,12] and frequency-shifting antenna[3,13]. Commonly used materials for 4D printing are soft composites (elastic modulus range $10^{-4} – 10$ MPa), such as

hydrogels[2,4,14,15], poly(dimethylsiloxane) (PDMS)[3,16,17], liquid crystal elastomers (LCEs)[12,18–21], and shape memory polymers (SMPs)[22–24]. However, the low elastic modulus ($E \leq 10$) of these materials (Table S1), when undergoing shape change, limits the scalability[3], actuation stress[14], and load-bearing capabilities of current applications[1].

Recently, we addressed this challenge by introducing multiscale heterogeneous epoxy composites (MHECs)[5], achieving a remarkable $10^4$ increase in $E$ compared to previous efforts (see Table S1), while demonstrating reversible shape changes comparable to state-of-the-art materials[3,4]. The mechanism of these actuators is governed by Timoshenko's theory for bimetallic thermostats, wherein a bilayer composed of two stiff materials with differing coefficients of thermal expansion (CTEs) undergoes curvature change due to thermal mismatch. This CTE mismatch induces bending upon Joule heating, resulting in a predictable and programmable shape transformation. Depending on the application, individual bilayers can operate as standalone actuators or be arranged into systematic lattice configurations to enable more complex, multi-curvature shape changes. Notably, this study showcased added functionalities, including addressable and localized Joule heating actuation, alongside self-

[1]Boston University Department of Mechanical Engineering, 110 Cummington Mall, Boston, MA, USA. [2]Boston University Division of Materials Science and Engineering, 15 St Mary's St, Boston, MA, USA. ✉e-mail: jwboley@bu.edu

sensing capabilities. These capabilities stem from the increase in electrical conductivity ($\sigma$) of the utilized materials composites, marking a notable advancement in the multifunctionality compared to others[5]. While the high $E$ contributes to increased actuation stress, load-bearing capabilities, and scalability, these materials are predominantly constrained by their inherent brittleness. Consequently, structures constructed from these materials exhibit susceptibility to mechanical failure even under minor strain perturbations, underscoring their inherent limitation in terms of damage tolerance.

One method to quantify this is by characterizing the toughness ($K$) of these materials, defined as the energy per unit volume needed to induce mechanical failure[25]. A summary of $E$ and $K$ values for commonly used 4D printable materials is reported in Table S1. Current 4D printable materials exhibit a trade-off between failure strain and $E$. Materials with a high failure strain are flexible and stretchable but tend to have a low $E$, while those with a high $E$ usually have a low failure strain, making them more brittle. Consequently, an existing challenge in 4D printing is the development of materials capable of maintaining a large, predictable, and addressable morphing mechanism for intricate shape transformations while enhancing both $E$ and $K$ for high-performance applications.

To address these challenges, we introduce a synthetic muscle composite to enhance the $E$ and $K$ of our 4D printed structures, all while maintaining a large, predictable, and addressable actuation mechanism. Inspired by the skeletal muscle system[26,27], we combine PDMS and MHEC to develop a synthetic analog. In the skeletal muscle system, fascia tissue plays a crucial role in maintaining muscle fiber integrity, akin to an exoskeleton[27]. Additionally, fascia tissue facilitates neural connections involved in sensing mechanical changes and pain perception[28]. The synthetic muscle composite exhibits significant similarities to natural skeletal muscles, utilizing high elastic modulus MHEC inks[5] as a synthetic muscle for mechanical force generation, and PDMS acting as synthetic fascia (SF), binding the composite and enabling electrical connections even after mechanical failure (see Fig. 1a). The MHEC inks constituting the synthetic muscle composite are developed in our previous work[5]. The synthetic fascia inks are composed of a PDMS matrix with adjustable cross-link density, displaying strong adhesion to MHECs. Within this framework, we developed five SF inks for DIW, allowing further tuning of the interfacial adhesive strength ($P$) (10 to 300 N m$^{-1}$) and $K$ (4 to 6 MJ m$^{-3}$). Through the integration of these soft adhesive materials with high-stiffness thermally responsive MHECs via DIW, we achieved an improvement of about three orders of magnitude in the $K$ of the resulting synthetic muscle composite (~27 MJ m$^{-3}$) compared to MHECs (~0.3 MJ m$^{-3}$), all while maintaining high stiffness (~22 GPa) and morphing mechanism. Utilizing this fabrication method, we printed an electrically controllable bilayer exhibiting damage tolerance and detection, enduring up to 7 fractures while continuing to function as programmed. Furthermore, we integrated the synthetic muscle composite into a previously designed lifting robot[5], setting records in lifting capabilities (~1230 times its own weight) and actuation stress (5.8 MPa) when compared to other 3D printed actuators. Notably, even after failure, the actuator maintained its operational integrity and high performance due to the high $K$ imparted by the synthetic fascia. Ultimately, we present a 4D printed lattice structure featuring the incorporation of the synthetic muscle composite, showcasing a sensitive electrically responsive surface with fracture detection capabilities. To emphasize this, we subjected one of these 4D printed lattices to extreme conditions, driving a car over it. Notably, the lattice structure detected fractures and exhibited high resilience, enduring external compressive damage equivalent to ~$3 \times 10^5$ its own weight.

## Results

### Synthetic fascia enables tough, stiff muscle composites
The formulation of the inks commences with a poly(dimethylsiloxane) (PDMS) resin exhibiting an operational temperature range of −54 to 290 °C, and a high elongation (150–250%)[29]. This PDMS was chosen since it can be polymerized at high temperatures (up to 290 °C), and can bond to a variety of substrates[29]. To develop inks suitable for direct ink writing (DIW) (i.e., a shear-yielding stress, shear-thinning response, and plateau storage modulus[30]), we included fumed silica (FS) into the PDMS formulations, serving as rheological modifiers[3,5] (see Fig. S3). As means for tuning $P$, $E$, and $K$, we vary the base-to-cross-linker weight ratio within the PDMS matrix.

Based on prior work on high toughness adhesives[31], we reason that the high adhesion (i.e., peel strength $P$) of our SF strongly correlates with high toughness synthetic muscle composites. In this regard, we systematically characterize $P$ and $K$ between one of our MHEC inks and various base:crosslinker formulations of our SF inks (Fig. 1b, c). Indeed, we observe an increase in necking of the interfacial SF with increased base:crosslinker (Fig. 1b) and corresponding increase in $P$, which has a strong positive correlation with $K$.

This high adhesion that we observed between our SF and our MHEC is uncommon for PDMS. Commercial PDMS manufacturers can increase the adhesion of non-polar PDMS polymers using a combination of fillers[32] and silane coupling agents[33] (see Fig. S1). To investigate the origin of this high $P$ we perform a systematic study comparing SF to other commercial PDMS formulations, including Ecoflex 00–30 and SE1700. We found that the $P$ for SF (~300 N.m$^{-1}$) is 3 orders of magnitude higher than both Ecoflex 00–30 (~0.30 N.m$^{-1}$) and SE1700 (~0.46 N.m$^{-1}$) (Fig. S6). We then conduct chemical analysis to further understand the difference between our SF and the other commercially available PDMS. When compared to other commercial PDMS formulations, the SF was found to have nearly undetectable quantities of non-polar Si-H groups (Fig. S2), which are known to significantly reduce adhesion between PDMS and a variety of substrates[34]. We also find an ingredient that is unique to our SF, namely 2-ethyl-1-hexanol (Fig. S2), which can perform oxirane ring opening reactions[35–37], resulting in strong chemical bonds between the SF and the MHEC. Notably, there may be other fillers and coupling agents in the SF that help contribute to the high $P$, which are not detected in our analysis (Fig. S6). However, the absence of the Si-H and the presence of 2-ethyl-1-hexanol are the strongest known contributors to the high degree of adhesion that we observe between the SF and MHECs. We also observed that the printing process further increases the adhesion between the SF and the MHEC. Compared to cast and co-cured SF-MHEC laminates, the ones that were co-printed and co-cured exhibit an ~8 times higher peel strength. This is likely due to a significant increase in polymer chain mixing and inter-layer diffusion[38,39], in the printed samples as the printed interfacial filaments overlap by ~5%.

Figure 1d shows the stress-strain relationship for the neat MHEC, the neat SF, and the synthetic muscle composite to illustrate the synergistic mechanical effect when wrapping the MHEC with the SF. The MHEC alone exhibits a high $E$ (~31 GPa), but a very low strain to failure (~0.3%). Conversely, the SF by itself exhibits a low $E$ (~2.5 MPa) and a relatively high failure strain (~344%). However, when the MHEC is wrapped with the SF, the resulting synthetic muscle composite preserves a high $E$ (~22.41 GPa) that is comparable to the underlying MHEC, with a large yield region, and a high failure strain (~187%) on the same order of magnitude as the SF. This synthetic muscle composite represents a class of 3D printed actuators that exhibit a high $E$ (similar to MHEC) and a $K$ that is higher than any other 3D printed actuator to date (~3 orders of magnitude higher than MHECs alone and ~3x higher than the toughest soft actuator) (Fig. 1e).

We can further tune $E$ and $K$ of these synthetic muscle composites through variations in the MHEC-to-SF ratio within the samples (see Fig. S7). As illustrated in Fig. 2a, a schematic outlines the manufacturing approach employed to adjust $E$ and $K$ based on the volume fraction of the constituent materials. Representative photographs of distinct cross sections corresponding to varying proportions of MHEC and SF are presented in Fig. 2b. Figure 2c, d reports the effective $E$ and $K$ as

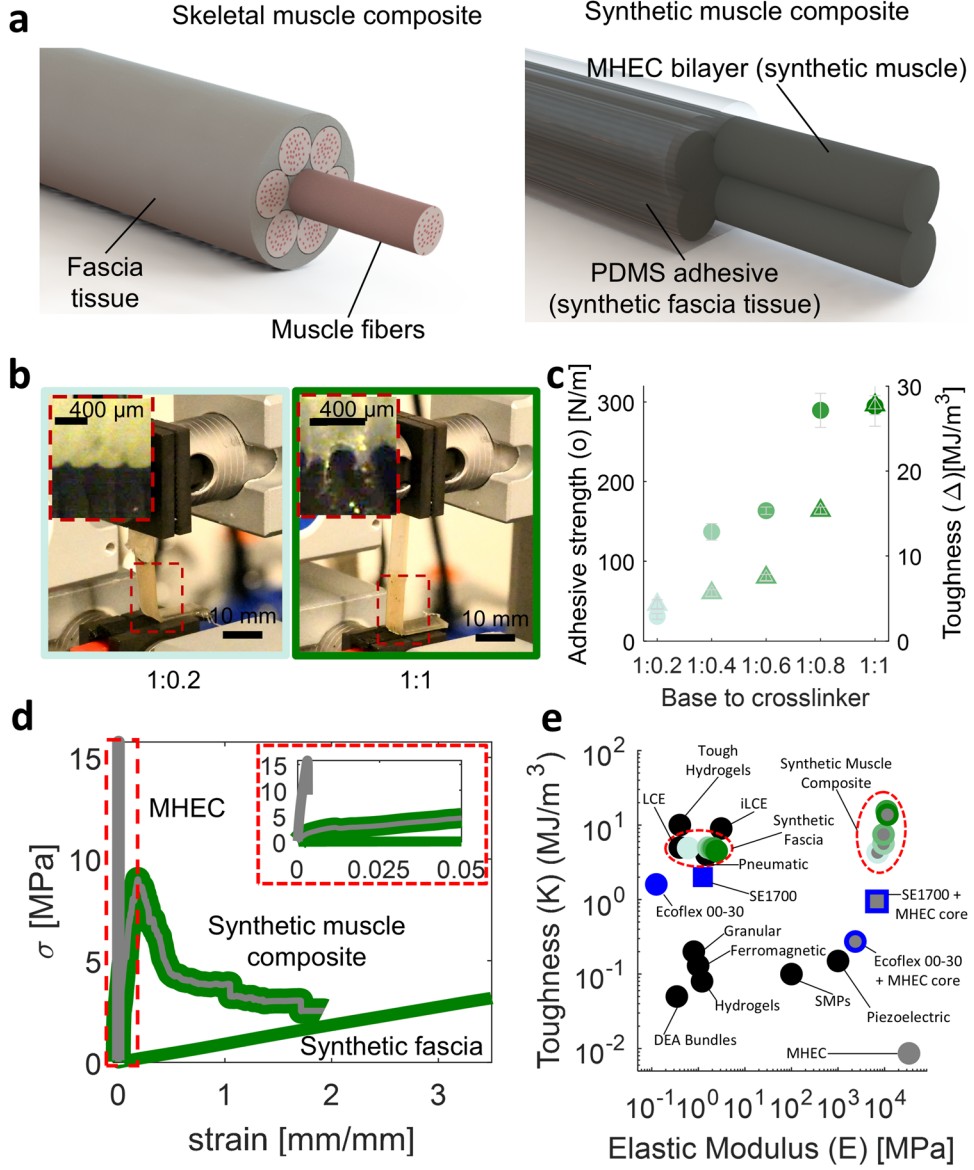

**Fig. 1 | Printable synthetic muscle composites. a** Representative schematic of human skeletal muscle composite, composed of muscle fibers encapsulated by a thin, soft tissue known as fascia. (left). Representative schematic of the synthetic muscle composite developed in this study (right). **b** Representative photographs of the peel testing (left). Insets represent close ups of the testing to illustrate tunable adhesive necking between the PDMS and MHEC. **c** Resulting peel strength (*P*) and toughness (*K*)as function of different base to crosslinkers formulations. **d** Representative stress-strain curve of constituent's materials, and the synthetic muscle composite. Inset on the top-right is used to focus on the stress-strain response of the MHEC sample. **e** Range of reported toughness and elastic modulus for different 3D printed actuators materials. Black data points represent previous work on printable materials. Blue data points represent non-stick PDMS formulations tested in this work. Green data points represent high-adhesive PDMS (SF) formulation presented in this work. Data points with gray cores represent different synthetic muscle composites tested in this work. MHEC ink used in this study is low $\alpha$ - SUP 1:0.2 + 15% v/v CF + 8% v/v CB (electrically conductive)[5].

functions of the different volume fractions between MHEC and SF (see Fig. S8). As expected, *E* exhibits a discernible trend wherein its values increase with a higher content of MHEC in the composite (see Fig. 2c). The contributions of the SF and MHECs to the resulting effective *E* are slightly lower than predicted by the rule of mixture model. This can be attributed to the interfacial bonding between phases, which may be influenced by the roughness of the materials. The roughness can generate stress concentrations, leading to a reduction in $E$[40,41]. Furthermore, the results for *K* reveal an interesting observation: the optimal proportions for maximizing toughness are approximately 80 % v/v MHEC and 20 % v/v SF (see Fig. 2d). Beyond this composition, specifically exceeding 80 % v/v of MHEC, the composite begins to deviate from its ductile-like responses and transitions toward a more brittle behavior, aligning closely with typical characteristics of pure

MHECs. This optimal combination, resulting in the highest *K*, can be attributed to the mitigation of crack propagation due to the combination of the SF "soft" phase and the MHEC "stiff" phase. These findings are further supported by prior related work on non-responsive stiff and tough printable composites[42].

## Self-sensing, damage-tolerant bilayer actuators

To showcase the practicality of our responsive synthetic muscle composite, we designed and 4D-printed an electrically controllable MHEC bilayer, encapsulated within an SF (Fig. 3a, top) (see Movie M1). The SF functions as soft "tissue" that will maintain the geometrical integrity of the bilayer, hence, acting as a soft ectoskeleton Fig. 1a[26–28]. For the core MHEC bilayer, we selected a combination of two stiff epoxy-based inks with distinct CTEs. The mismatch in CTEs enables

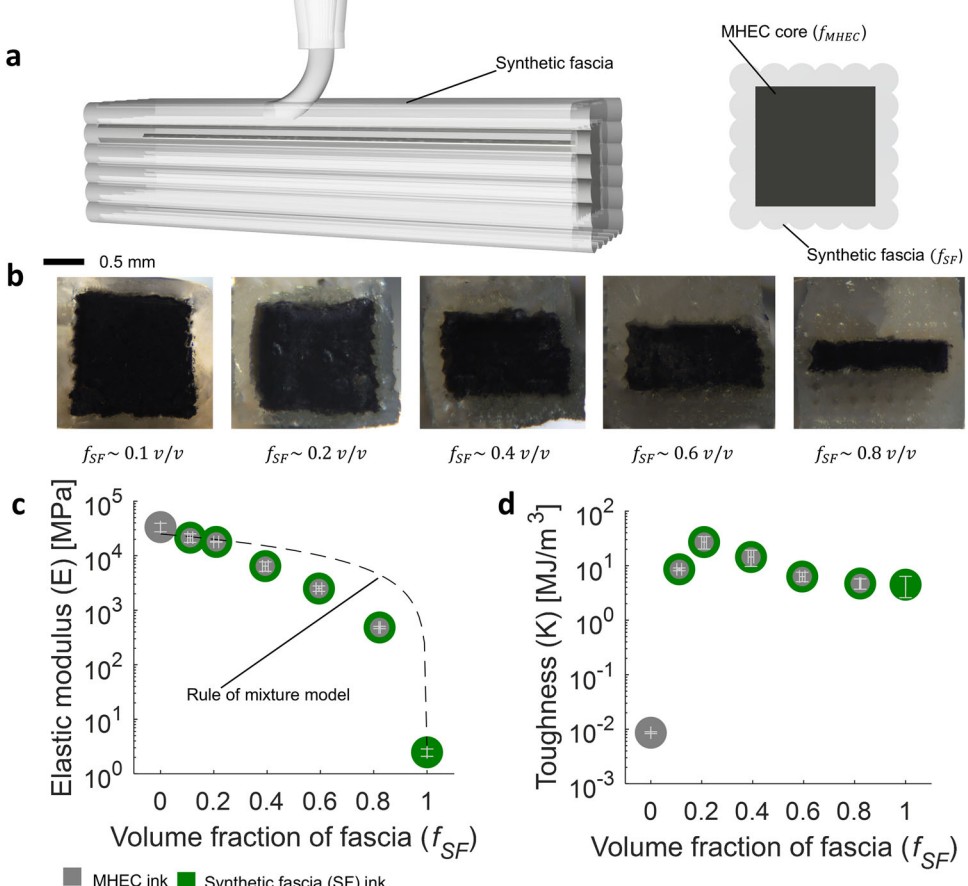

**Fig. 2 | Synthetic muscle composite with tunable $E$ and $K$. a** Printing schematic of the synthetic muscle composite (left). Schematic of the cross-section of the composite, illustrating $f_{MHEC}$ and $f_{SF}$, where they represent the volume fractions of MHEC and SF, respectively. **b** Representative photographs of different $f_{MHEC}$ and $f_{SF}$ tested. **c** Summary of the results for $E$ with respect to different $f_{SF}$. Dotted line represents the rule of mixture model (see SI Methods). **d** Summary of the results for $K$ with respect to different $f_{SF}$. SF ink used in this study is MS 1:1 + 2.8% v/v FS. MHEC ink used in this study is low $\alpha$ - SUP 1:0.2 + 15% v/v CF + 8% v/v CB (electrically conductive)[5].

curvature changes upon heating, as described by Timoshenko's theory for bimetallic thermostats. To enable active control via Joule heating, at least one of the inks must be electrically conductive. This allows us to locally and precisely regulate temperature through electrical input. For additional details on the design and composition of the electrically responsive MHEC bilayer, we refer the reader to our previous work[5]. For the SF ink, we selected one that demonstrates superior adhesion to MHEC (MS 1:1 + 2.8% v/v FS), translating to a higher $K$ (see Fig. 1). To evaluate the performance of the synthetic muscle composite, we induced equidistant fractures along the arc length of the bilayer, as depicted in Fig. 3a (bottom). Figure 3b presents the change in electrical resistance ($R/R_o$) across the bilayer composite over time, revealing discernible fractures and highlighting its inherent self-fracture detection properties. These fractures are manually induced on a single bilayer sample in a sequential manner. Each set of peaks in the signal corresponds to a new fracture event. At lower peaks (from $N = 1$ and $N = 3$), we observe a single peak for each new fracture, with a magnitude increasing with $N$. From $N = 4$ to $N = 5$ we see a single high magnitude primary peak, likely resulting from the new fracture, and the presence of additional secondary peaks, which is likely a result of small motions at the interfaces of the existing fractures. Above $N = 5$ we see multiple peaks of similar magnitude, which is likely a result of the additional fracture and collective interfacial slipping across the multiple existing fractures of the sample. Moreover, we sought to assess whether the bilayer composite could sustain actuation responses post-fracture. We subjected the bilayer composite to actuation, transitioning from room temperature (0 W) to approximately 110 °C (~ 6.57 W). In Fig. 3c, the actuation response is illustrated for scenarios with no fractures ($N = 0$) (top), three fractures ($N = 3$) (middle), and seven fractures ($N = 7$) (bottom), demonstrating consistent modulated actuation even after multiple fractures (see Movie M2). Figure 3d further supports this by revealing localized heat generation near fracture points (see Movie M3), attributed to slightly higher electrical resistance in those areas due to the change in contact area (e.g., reduction in contact area and increase in contact roughness[43,44]). Inspired by previous work[5,12], we implemented an electro-thermomechanical model, taking into account the differences in materials and dimensions in this system, which mainly will affect the thermal response time[45] (see SI Methods: Electro-Thermo-Mechanical Modeling Details). Figure 3e, f demonstrates good agreement between the model and experimental electrical ($\Delta R/R_o$) and curvature ($\delta\kappa$) responses, respectively, as a function of induced power, affirming that the bilayer composite retains key features such as self-sensing, repeatability, reversibility, and modulated actuation through Joule heating, a feat attributed to the addition of the synthetic fascia. Moreover, based on our previous work demonstrating stable performance over more than 20,000 actuation cycles for the underlying MHEC system, and considering the electrically and mechanically decoupled nature of the synthetic fascia, we expect comparable or improved long-term durability for the integrated structure. Notably, it also exhibits additional capabilities such as fracture detection and tolerance.

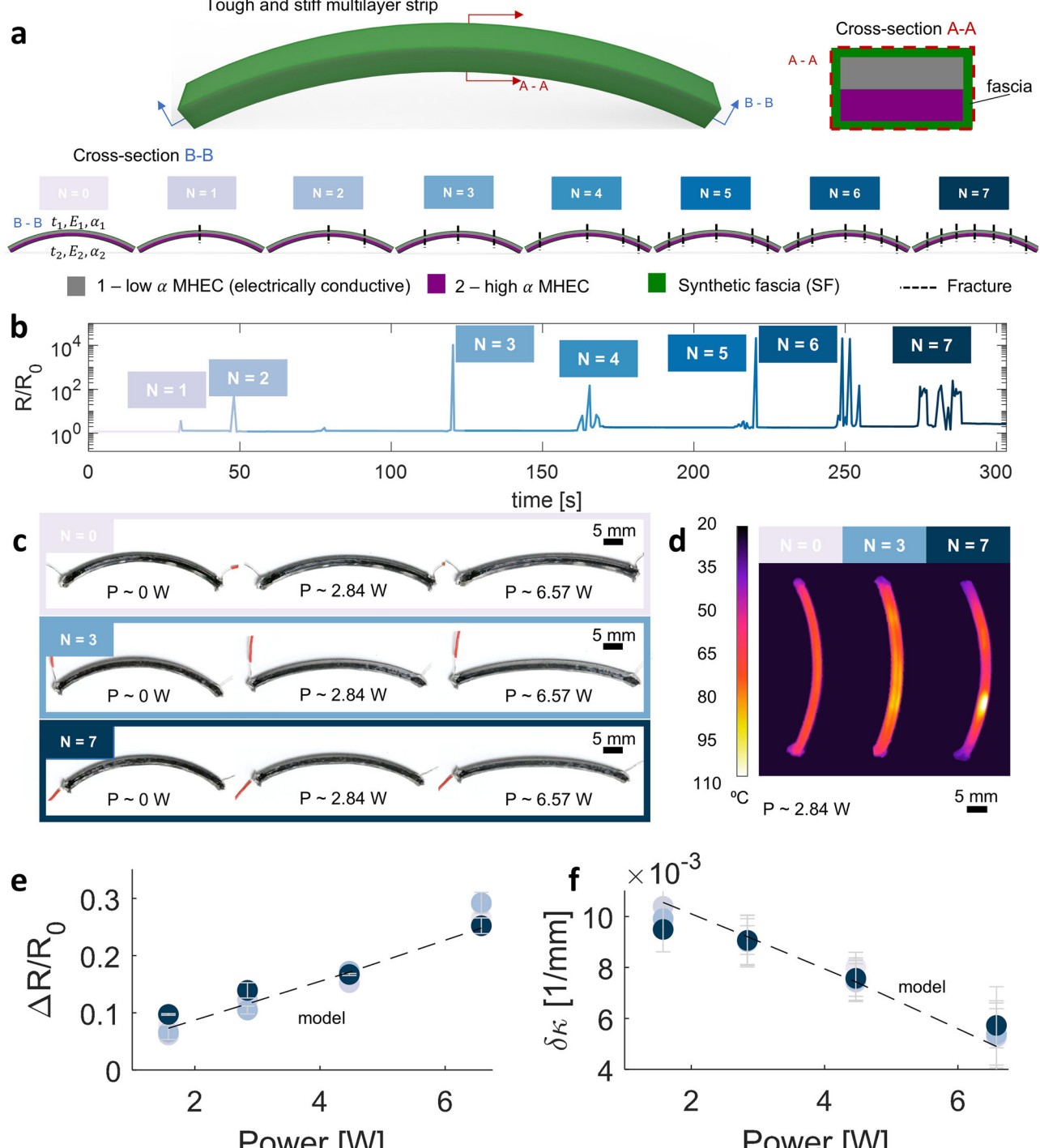

**Fig. 3 | Multimaterial 4D printed electrically controllable bilayers with damage tolerance and detection. a** Schematic of the synthetic muscle composite bilayers (top-left). Schematic of the cross-section (A-A) of the bilayer, showing the integrated materials (top-right). Schematic of the cross-section (B-B) of the bilayer, showing the locations of induced fractures ($N$) along the arc length (bottom), ranging from 0 to 7. Different tones of blue indicate the number of induced fractures ($N$). MHEC inks used in this study are low $\alpha$ - SUP 1:0.2 + 15% v/v CF + 8% v/v CB (electrically conductive) and high $\alpha$ - SUP 1:1.8 + 24% v/v surfactant + 8% v/v FS (electrically insulative). SF ink used in this study is MS 1:1 + 2.8% v/v FS. **b** Measured change in electrical response ($R/R_o$) with respect to induced fractures ($N$), across time. **c** Optical images of electrically controllable actuation via Joule heating at different power levels, for samples with no fractures ($N = 0$, top), samples with three fractures ($N = 3$, middle), and samples with seven fractures ($N = 7$, bottom). **d** Thermal imaging corresponding to samples tested via Joule heating, with respect to the number of fractures ($N = 0$, 3, and 7). **e** Measured electrical ($R/R_o$) responses (filled circles) with respect to different power levels. **f** Measured curvature ($\delta\kappa$) responses (filled circles) with respect to different power levels. Different tones of blue represent different numbers of fractures ($N$) on the sample. Dotted lines represent the electro-thermomechanical model (see SI Methods).

## Damage-tolerant heavy lifting robot

To illustrate the capabilities of our synthetic muscle composite, we constructed a shape-shifting weightlifting robot using 4D multi-material printing (Fig. 4a). The design is based on insights derived from our previous work, 5 distinguished by the incorporation of SF27, which envelops the MHEC structure (see Movie M4). This integration results in the creation of a stiff and tough lifting robot. The SF ink employed is MS 1:1 + 2.8% v/v FS, selected for its high $K$ (see Fig. 1). To optimize the actuation performance, we focused on enhancing both $K$ and $E$ for this composite structure, essentially striving for a MHEC-to-SF volumetric ratio close to the optimal configuration (80 to 20) as depicted in Fig. 2. Considering the requirements of the weightlifting robot's design depicted in previous work[5], we achieved a MHEC-to-SF volumetric ratio of 7o to 30.

Figure 4b illustrates the evaluation of the stiff and tough lifting robot's performance. We initiated the assessment by subjecting the robot to modulated actuation using Joule heating. The actuation cycle commenced with a fixed voltage input of 30 V, with the current limited to a maximum of 0.2 A, resulting in an approximate power output of 6 W sustained for approximately 25 s (approximately 1 time constant, refer to SI Methods: Electro-Thermo-Mechanical Modeling Details). This was succeeded by a power cut-off and a cooldown interval of about ~ 120 s (approximately 5 time constants, refer to SI Methods: Electro-Thermo-Mechanical Modeling Details). Subsequently, external weight (a combination of a glass slide and/or calibrated weights) was incrementally added to the robot, and performance metrics were tracked until reaching mechanical failure. The monitored actuation metrics included specific output work, lifted specific mass, and stroke. The representative image in the top panel of Fig. 4b depicts the testing setup before reaching failure. The center panel of Fig. 4b illustrates the point of failure, marked by one of the actuator legs snapping in two under the high external loads being carried while undergoing actuation. However, the incorporation of the SF enabled the retention of some structural integrity, allowing the robot to remain functional to some extent (Fig. 4b bottom panel). This indicates that we now have a structure capable of operating under extreme conditions (Fig. 4b top panel and Movie M5), experiencing mechanical failure (Fig. 4b center panel), and still being able to actuate afterwards (Fig. 4b bottom panel and Movie M6). In Fig. 4c, d, the measured performance of these actuators is shown with respect to lifted mass for different testings, where blue, yellow, and black dots represent the lifting robot before, after, and right at the mechanical failure, respectively. The stiff and tough lifting robot demonstrated a maximum specific work of 14.64 J kg$^{-1}$ and an actuation stroke of 18.52%, before reaching mechanical failure. It was able to lift a maximum of 1231 times its own weight, which is approximately 7.89% higher than the strongest natural example, the dung beetle, capable of lifting 1141 times its own weight[46,47]. Conversely, after reaching mechanical failure, the lifting robot continued to function, yielding a maximum specific work and actuation stroke of 6.31 J kg$^{-1}$ and 15.67%, respectively, while lifting a maximum of 564 times its own weight. In contrast to prior work, these structures exhibited high performance in a self-standing configuration and maintained comparable performance even when structural integrity was compromised.

Figure 4e presents actuation stress and specific mass for relevant previous works on printable responsive actuators[4,12,16,20,22,48–50]. Actuation stress is defined as the lifted weight divided by the cross-sectional area of the actuator. Following a classification similar to prior studies[5], we categorized the actuators into two groups based on their actuation mechanisms: via change in environment conditions through additional components (passive) (e.g.'s, hot plates, ovens, external magnets, water bath) or using internal stimuli (active) (e.g., voltage). Observing the results, the stiff and tough lifting robot before mechanical failure (indicated by a blue star marker) surpasses our previously set records (MHECs), showcasing improvements of ~ 4% and ~ 39% in actuation stress and specific mass, respectively. In contrast, post-mechanical failure (indicated by the yellow star marker), the stiff and tough lifting robot experiences a decrease in actuation stress and specific mass of ~ 52% and~ 36%, respectively, compared to our previous work (MHECs). Nevertheless, compared to other actuators, it continues to outperform both active and passive previous works in terms of generated actuation stress, while achieving the highest lifting capacity among active actuators. Furthermore, a comparison of the $E$ and $K$ of various materials used in printable actuators reveals that the synthetic muscle composite employed in this stiff and tough lifting actuator exhibits enhancements ~ 71% on $K$, while maintaining comparable high $E$ to our previous work (MHEC)[5] (see Fig. S9 and Table S2). Similarly, when compared to commercially available options, our synthetic muscle composite ranks as the third-highest performing material (in terms of $E$ and $K$) among 10 different options (see Figs. S10, 11 and Table S3).

## 3D Sensing Surface that Detects and Tolerates Damage

Inspired by the improvements demonstrated by the stiff and tough lifting robot, we aimed to incorporate the SF into intricate electrically responsive shape-shifting lattices, thus constructing a stiff and tough electrically responsive lattice (see Movie M7). The general design of the lattice is detailed in our prior work[5], and the specifics of the design uses in this work is detailed in section. The integration of the SF layer was seamlessly embedded within the multi-material 4D printing manufacturing method, as depicted in Fig. 5a. The SF ink used was MS 1:1 + 2.8% v/v FS, chosen for its highest $K$ (see Fig. 1). The MHEC inks used to construct this morphing electrically controlled lattice where developed in previous research[5]. Considering the requirements of the lattices design depicted in previous work[5], we achieved a MHEC-to-SF volumetric ratio 51 to 49. To assess the resilience of these electrically controlled morphing lattices, we devised an experiment exposing the lattice structure to external perturbations, including compressive cycling loadings (i.e., single-axis mechanical tester) and high-impact loadings (i.e., sledgehammer impacts). Figure 5b provides a visual representation of a complete cycle in the characterization process of these lattices. The cycle commenced with the application of compressive cycling forces (10 compressive cycles) onto the lattice surface. Simultaneously, we monitored the change in resistance ($R/R_o$) across the lattice over time. Subsequent to the compressive cycling, the lattice underwent external perturbations (e.g., hits with a sledgehammer), while still tracking the change in electrical resistances ($R/R_o$). This entire characterization cycle was repeated once for the MHEC lattice (Fig. S12) and six times for the synthetic muscle composite lattice (Fig. 5b).

Starting with the MHEC lattice characterization (Fig. S12 and Movie M8), it is noteworthy that the force generated in the initial compressive cycle was approximately three times higher than in subsequent cycles. This notable spike can be attributed to a sudden fracture in the lattice structure during the first cycle (see Movie M8), adversely impacting its structural integrity and subsequently reducing the required force for compression. Additionally, as external perturbations were applied, the structural integrity of the lattice was visibly compromised, leading to its inability to maintain its hemispherical form. This observation is further supported by the results of the electrical response, revealing a significant jump in the change in the electrical resistance ($R/R_o$) during external perturbations. This abrupt increase suggests that the ribs of the lattice were entirely disconnected, and the recorded resistance value indicated an open-loop condition. These findings are reinforced by supplementary Movie M8, where the induced fractures in the structure during testing can be visually observed.

Conversely, the synthetic muscle composite lattice (Fig. 5c and Movie M9) demonstrates remarkable structural resilience, sustaining up to six characterization cycles. Notably, it is observed that the force required to compress the lattice decreases with each subsequent

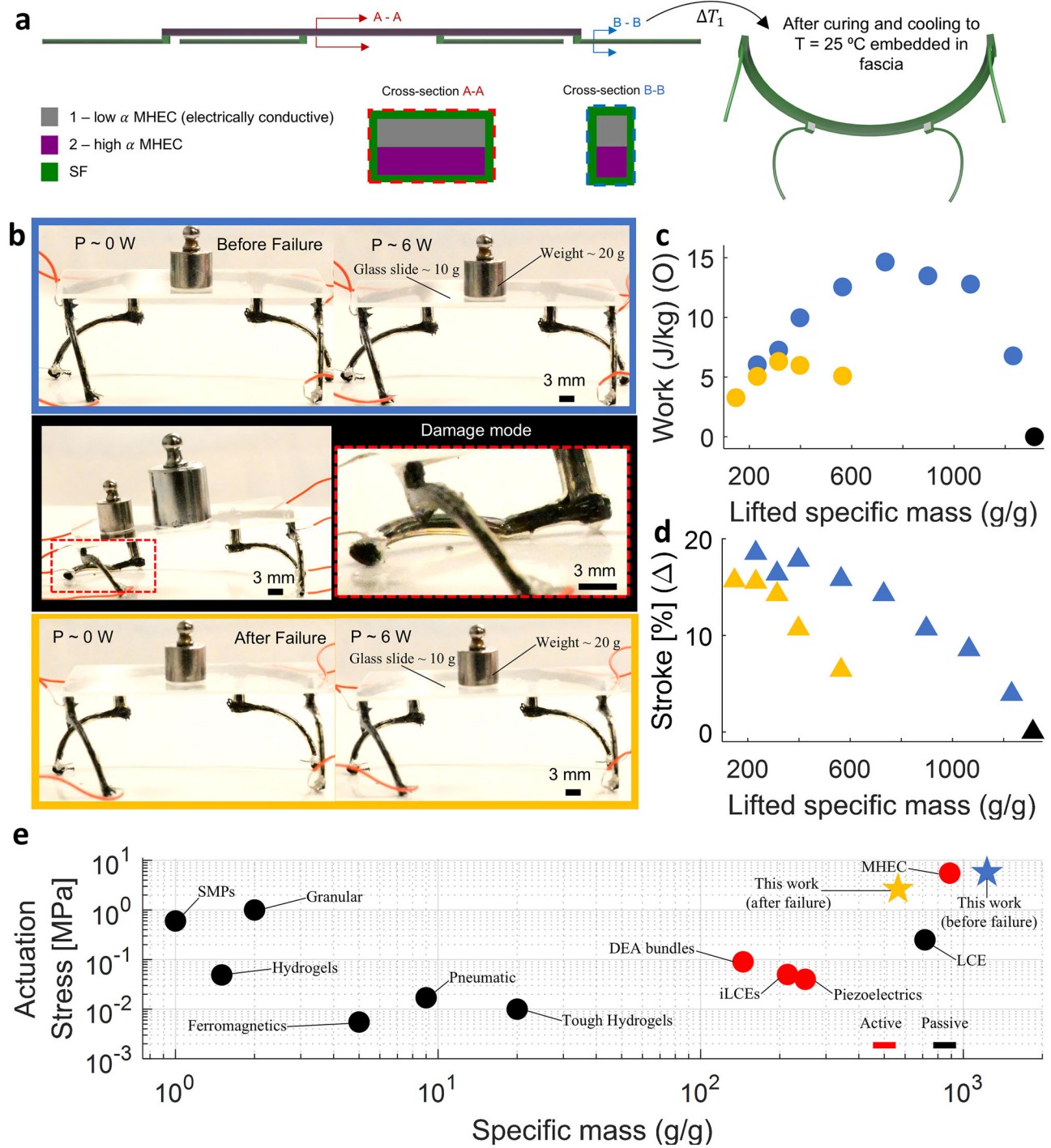

**Fig. 4 | Multimaterial 4D printed stiff, tough and damage tolerant weightlifting robot. a** Schematic of as printed lifting robot, alongside with cross sections, where A-A and B-B illustrate the composition of the connectors and legs, respectively (left). Schematic of as cured and cooldown to room temperature of the stiff and tough lifting robot (right). MHEC inks used in this study are low $\alpha$ - SUP 1:0.2 + 15% v/v CF + 8% v/v CB (electrically conductive) and high $\alpha$ - SUP 1:1.8 + 24% v/v surfactant + 8% v/v FS (electrically insulative). SF ink used in this study is MS 1:1 + 2.8% v/v FS. **b** Photographs of the lifting robot being actuated before mechanical failure, lifting 532 times its own weight (top–blue). Photographs of the damage mode of the lifting robot, inset represents a closed up of the damage mode (middle–black). Photographs of the lifting robot being actuated after mechanical failure, lifting 532 times its own weight (bottom - yellow). **c** Measured specific output work with respect to specific lifted mass for different lifting tests, where blue, yellow, and black dots represent the lifting robot before, after, and right at the mechanical failure, respectively. **d** Measured actuation stroke with respect to specific lifted mass for different lifting tests, where blue, yellow, and black dots represent the lifting robot before, after, and right at the mechanical failure, respectively. **e** Performance metrics of different 3D printed actuators, where black are passive actuators, red are active actuators, and blue and yellow stars represent this work before and after failure, respectively.

external perturbation (e.g., hits with a sledgehammer), indicating the introduction of fractures in the lattice structure. This observation aligns with the spikes in the change in electrical resistance ($R/R_o$), coinciding with the application of external perturbations, showcasing an inherent fracture detection capability. An intriguing behavior is evident in the electrical responses of the lattice: as additional fractures are induced, the sensitivity of the resistance in response to deformation increases. This phenomenon is attributed to the amplified change

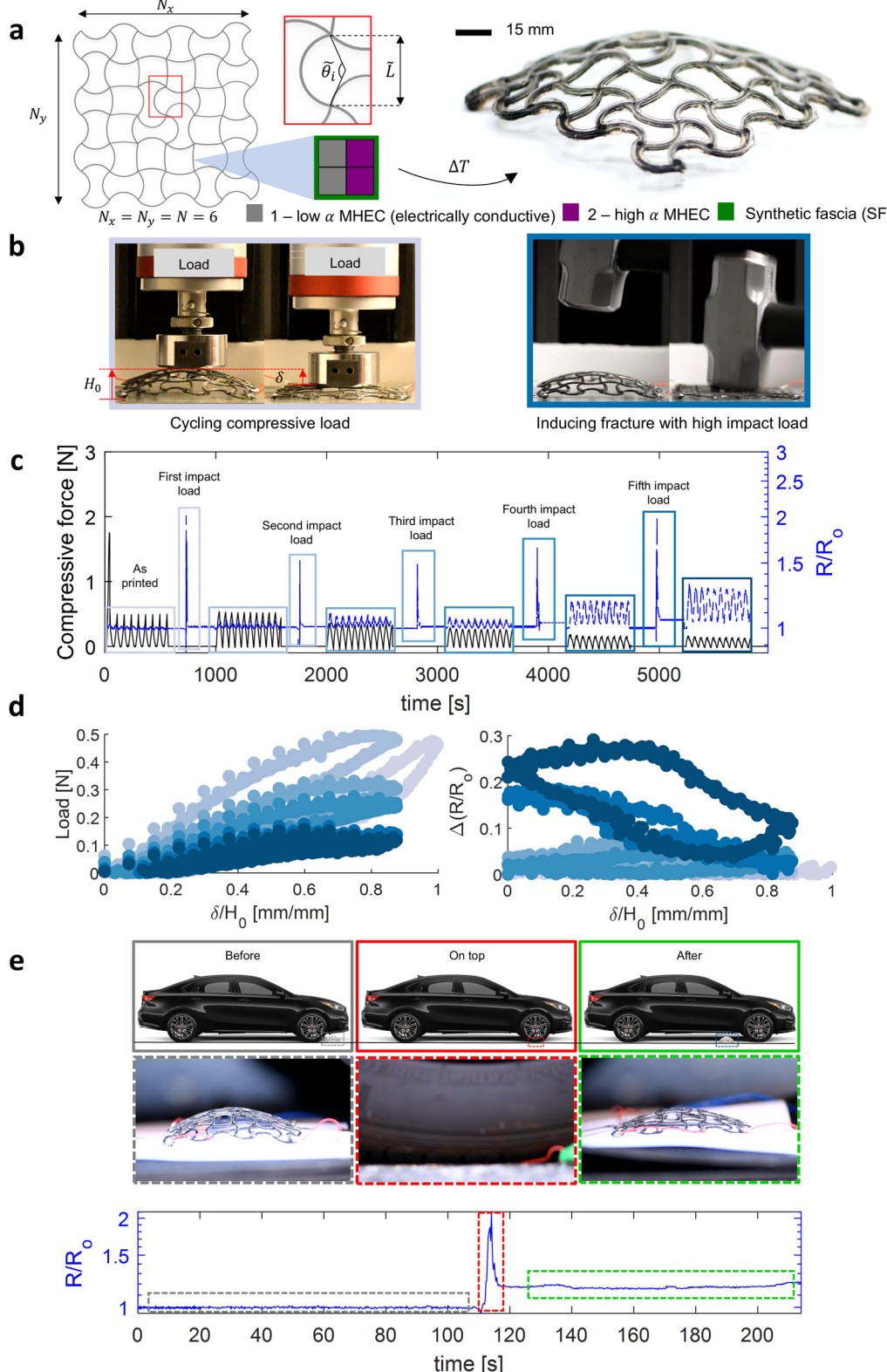

in resistance resulting from slight relative movements between segments of the fractured lattice ribs. These movements reduce the contact area within the fractured ribs, thus increasing the electrical resistance with respect to an external force. However, since the synthetic fascia surrounding the electrically conductive ribs is hyperelastic and adheres to the MHEC ribs, after the compressive loads are released, the MHEC ribs can revert to their initial configuration,

increasing the contact area of the ribs and thereby reducing the electrical resistance of the lattice close to its nominal value. Prior research on soft sensors composed of conductive fillers provides additional support for these findings[51].

A summary of these findings is presented in Fig. 5d. The synthetic muscle composite lattice exhibits varied force and resistance responses, depending on the number of induced fractures (i.e., the number of

**Fig. 5 | Multimaterial 4D printing of stiff and tough 3D sensing lattice.**
**a** Schematic designing and printing process of the composite lattice (left). Photograph of the composite lattice as printed and cooldown to room temperature (right). MHEC inks used in this study are low $\alpha \cdot$ SUP 1:0.2 + 15% v/v CF + 8% v/v CB (electrically conductive) and high $\alpha \cdot$ SUP 1:1.8 + 24% v/v surfactant + 8% v/v FS (electrically insulative). SF ink used in this study is MS 1:1 + 2.8% v/v FS.
**b** Experimental setup for external compression cycling (left) and impacts (i.e., sleedhammer) (right) exerted on the lattice surface. **c** Measured force (black solid line) and electrical response ($R/R_o$) (blue dotted line) of the lattice, while subject to external perturbations (i.e., compressive and impact). Insets represent different

stage from the testing, from no fracture (as printed), up to the highest amount of induced fractures in this study (fifth impact load). **d** A summary of the results of the characterize lattices is presented, where each color represent a different stage on the testing. **e** Schematic of the experimental setup, where the lattice is subjected to high compression loading exerted by a car (top). Insets represent different instants during testing, specifically gray, red, and green represent before compression, compressed, and after compression, respectively. Electrical ($R/R_o$) response as a function of time, where different dotted-line rectangles correspond to the different stages of the testing (bottom).

external perturbations from the sledgehammer). At a low number of external perturbations (e.g., three or less), the surface exhibits a relatively high compressive force and low electrical responsiveness. At a higher number of external perturbations (e.g., four or more), the surface exhibits a low compressive force and high electrical response. Furthermore, this synthetic muscle composite demonstrates high resilience to various external perturbations, showcasing 3D surfaces capable of sensing and damage detection.

To further evaluate the resiliency, sensing, and damage detection capabilities, we subjected the synthetic muscle composite lattices to an extreme condition-driving a car over it (see Fig. 5e top and Movie M10). Figure 5e (bottom) displays the change in the electrical resistances ($R/R_o$) measured during the testing, accompanied by representative photographs illustrating different stages of the experiment. Notably, it can be observed that when the car is positioned completely over the lattice, there is a spike in the $R/R_o$, signifying the induction of fractures. Subsequently, there is a decline in $R/R_o$, accompanied by a permanent drift in the nominal reading of approximately ~15%, indicating a lasting change. This may be attributed to the permanent reduction in the contact area between electrically conductive ribs caused by the extreme loads[43,44,51]. It is crucial to emphasize that the lattice demonstrated remarkable resiliency during this extreme testing, withstanding compression under an external load equivalent to $3.3 \times 10^5$ times its own weight, and even after such significant stress, it retained its shape and actuation capabilities (see Movie M11).

## Discussion

We have developed a responsive synthetic muscle composite for 4D printing, showcasing enhanced $K$ (~ 3 orders of magnitude compared to MHEC and *sim* 3x higher than the toughest soft actuators), while maintaining high $E$ comparable to MHEC (*sim* 22.41 GPa). Furthermore, we achieve high $P$ with the combination of the PDMS chemistry and the multi-material co-printing and co-curing approach. We have demonstrated the ability to finely tune $E$ and $K$ with respect to the volume fraction of the SF and MHEC into the composite. Resulting in an optimal composition (maximum $K$, while maintaining high $E$) of 20 and 80% v/v of SF and MHEC, respectively. Furthermore, our work highlights the resilience of synthetic muscle composite bilayers, enduring up to seven fractures while retaining reversible, controllable, and predictable self-sensing actuation via Joule heating, all while including additional functionalities such as self-fracture detection and damage tolerance. Utilizing this synthetic muscle composite, we constructed a stiff and tough self-standing lifting actuator that sets records by lifting approximately 1230 times its own weight. Intriguingly, even after reaching mechanical failure, this lifting robot remains operational, surpassing the actuation stress of both active and passive printable actuators from previous works and achieving the highest lifting capacity among active actuators. Additionally, when benchmarked against commercially available options, our composite ranks as the third-highest performing material among 10 alternatives in terms of both $E$ and $K$. Lastly, we integrated the synthetic muscle composite into multi-material 4D printed electrically responsive lattice structures. These structures, constructed with the synthetic muscle composite, proved to be damage-tolerant and featured a sensitive electrically

responsive surface with fracture detection capabilities. In a rigorous test, we subjected one of these 4D printed lattices to extreme conditions by driving a car over it. Impressively, the lattice structure not only detected fractures but also exhibited high resilience, enduring external compressive damage equivalent to $3.3 \times 10^5$ times its own weight. Looking ahead, the potential integration of these materials into diverse structural designs promises the development of autonomous, stiffer, tougher, damage-detection, and responsive morphing systems-ranging from robots and sensors to antennas-ushering in a new era of versatile and resilient technologies.

## Methods

### Materials

All SF inks were created by mixing (DAC 150.1 FVX-K, FlackTek, 60 s at 2000 $r \min^{-1}$) appropriate amounts of base and cross-linker of PDMS (MasterSIL 323, Master Bond). Next, we added the appropriate amount of fumed silica (FS) (Aerosil 106, Evonik), ranging from 2.8–3% v/v and pre-mixing (DAC 150.1 FVX-K, FlackTek) for 360 s at 2000 $r \min^{-1}$ with cooling down steps of 60 s, every 120 s. After, we further mix (DAC 150.1 FVX-K, FlackTek) the inks for 90 s at 3500 $r \min^{-1}$ with cooling down steps of 30 s, every 30 s. The pot life of all these elastomer inks is about 3.5 h. All the inks are loaded into Luer-Lock syringes (3cc or 10cc, Nordson, EFD), then centrifuged (300 s at 4000 $r \min^{-1}$) to remove air bubbles before printing. A summary of ink formulations is listed in Table S4. Electrically conductive MHEC ink used in this work was created by adding the proper amount of corresponding base to cross-linker (SUP121AO, Master Bond), following with the addition of corresponding carbon fibers (K223HM, K223HM, Mitsubishi Chemical, diameter ~11 µm, length ~200 µm), and corresponding carbon black (100% compressed, 99.9%, Thermo Scientific Chemicals), and mixing (DAC 150.1 FVX-K, FlackTek) for 180 s at 3500 $r \min^{-1}$ with cooling down steps of 60 s, every 60 s[5].

Electrically insulated MHEC ink was created by mixing the required amount of surfactant (24% v/v) at 3500 RPM for 30 s (DAC 150.1 FVX-K, FlackTek), following by the addition of (8% v/v) fumed silica (Aerosil R 106, Evonik) and mixing (DAC 150.1 FVX-K, FlackTek) for 90 s at 3500 $r \min^{-1}$ with cooling down steps of 30 s, every 30 s. Eco flex 00–30 ink used in this work was created by separately mixing (DAC 150.1 FVX-K, FlackTek) fumed silica (5% v/v) (Aerosil 150, Evonik) with parts A and B, following a mixing schedule of 420 s at 3500 $r \min^{-1}$ with cooling steps of 60 s. Then mixing part A and B for 60 s at 1750 $r \min^{-1}$[52]. SE1700 ink used in this work was created by mixing the corresponding base with cross-linker (10:1), following the addition of 3.0% of fumed silica (Aerosil 150, Evonik), and mixing for 420 s at 3500 $r \min^{-1}$ with cooling steps of 60 s[3,53].

### Ink rheology

Rheological characterization is carried out using DHR $\cdot$ 2 rheometer (TA instruments), equipped with a 20 mm diameter plate geometry and gap size of 1.6 mm. Materials are equilibrated at room temperature for 30 s before flow and amplitude sweeps are conducted. In flow sweeps, the materials are sheared at 0.01 [$s^{-1}$] to 10 [$s^{-1}$]. Shear storage ($G'$) and loss storage ($G''$) moduli are measured as a function of shear stress at a frequency of 1 Hz during amplitude sweeps. Unlike the

control formulations, each ink displays clear plateau, yield stress, and shear thinning behavior, attributed to the addition of fumed silica. A summary of the rheological characterization of the inks is resented in Fig. S3.

## Adhesive fascia chemical characterization

Uncured PDMS (MasterSIL 323 (SF), Eco Flex 00–30 and SE1700) were prepared using their set mixing ratios (see Table S4), and mixed (DAC 150.1 FVX-K, FlackTek, 60 s at 3500 $r$ min$^{-1}$) and immediately taken for FTIR analysis. Cured PDMS samples were prepared the same way and cured (250 °C, 2 h). FTIR was taken from a neat version of the epoxy (Supreme 112SP, Masterbond), to prevent interference from carbon black and other fillers. The mixed uncured and cured samples of PDMS (MasterSIL 323, Eco flex 00–30 and SE1700 were analyzed using a Nicolet 4700 ATR-FTIR. Absorbance measurements were taken from 500–4000 cm$^{-1}$ with a step size of 1 cm$^{-1}$ and 128 scans.

Uncured PDMS components (1 g, MasterSIL 323 A, MasterSIL 323B, Eco Flex 00–30 A, Eco Flex 00–30 B, and SE1700 A and SE1700 B) were each sonicated in acetone (10 mL) for 20 min, and the solution was filtered (Thermo Scientific, Nalgene Syringe Filter 0.45 µm) to extract any polymer additives. Gas Chromatography Mass Spectrometry (GC-MS) was run on these solutions using an Agilent 6890N GC-MS in triplicate, and compared against the National Institute of Standards and Technology Mass Spectral Database (NIST 11)[54] to identify compounds.

## Multimaterial 4D printing of synthetic muscle composite

Samples are manufactured following methods and procedures stated in previous work[5], with the main differences that we included the new PDMS layer that encapsulated the MHEC sample. Extrusion printing pressures (7x 3cc HPx High-Pressure Dispensing Tool, Nordson EFD) and speeds ranged from 160–495 psi and 12–16 mm s$^{-1}$, respectively. Table S5 present printing parameters for each ink. The integration of the PDMS layer was achieved through a custom print path, combining MHEC inks and PDMS ink in a co-printing approach. During the incorporation of the PDMS layer, we strategically predefined exposed areas of the MHEC bilayer. This was done to facilitate electrical wiring connections for subsequent demonstrations.

## Peel strength, elastic modulus, and toughness measurements

Samples (8 mm wide, 50 mm long, and 3.3 mm tall) were printed to characterize the adhesive strength for each of our PDMS (SF) inks to MHEC (Fig. S4), following the standardized testing method ASTM D3330/D3330M - 0455[55] (see Movie M12). Each sample was tested in a single-axis mechanical tester (Instron 5944 Micro-tester) using a pulling speed of 0.2 mm s$^{-1}$ (Fig. S4). The resulting peel force versus peel length data is presented in Fig. S4. The peel/adhesive strength is calculated by taking the average peel force and then dividing this force by the width (~8 mm) of the adhesive sample. The average of the peel force is taken from 10 mm (peel length) to failure. A summary of the resulting peel/adhesive strength is given in Fig. 1b, c, with error bars representing the standard deviation of the measurements. Three samples were tested per ink formulation.

Test samples (gauge length 9.50 mm, wide 3.20 mm, and thickness 1.60 mm) were printed to characterize the elastic modulus of each PDMS (SF) ink and synthetic muscle composite, following standardized methods ASTM D0638 - 14[56] (see Movie M13). Each sample was tested under uniaxial tension in a single-axis mechanical tester (Instron 5944 Micro-tester). PDMS (SF) and synthetic muscle composite samples were tested at engineering strain rates of $2.8 \times 10^{-2}$ s$^{-1}$ and $1.6 \times 10^{-2}$ s$^{-1}$, respectively (Fig. S5). The $E$ for each sample was determined by fitting low-strain data ($0 \leq \epsilon \leq 0.005$)[57] to the linear relationship $\sigma = E\epsilon$ via the 'polyfit' function in MATLAB. The $K$ for each sample was determined by calculating the area under the curve of the stress versus strain data (Fig. S5). This was achieved using the numerical

integration 'trapz' function in MATLAB. A summary of the resulting $E$ and $K$ for each sample is given in Fig. 2c, d and TTable S6. Three samples were tested per ink formulation.

## Optimizing volume fractions for synthetic muscle composites

For the MHEC ink, we chose low $\alpha$ - 1:0.2 + 15% v/v CF + 8% v/v CB[5], due to its high electrical conductivity which allow us control of the structure using Joule heating, while also providing additional resistive sensing functionalities. For the SF ink we chose MS 1:1 + 2.8% v/v FS, which is the ink that exhibits the highest adherence to the MHEC inks (Fig. 1c). Synthetic muscle samples are developed following methods and procedures detailed in the section, with the consideration of varying the proportions of the SF and MHEC accordingly. The volume fraction of each constituent material in the composite was characterized by taking cross-sectional pictures of each sample using a calibrated miniature microscope lens system (InfiniTube FM-100, Infinity) and an 18 mm/2.00x lens (PL-series, Infinity). Data for the cross-section was extracted from the images using a custom MATLAB script. $E$ and $K$ were characterized following the methods and procedures stated in the section. Three samples were tested for different MHEC-to-SF proportions.

## Characterizing damage-tolerant electromechanical bilayers

Simple bilayers are constructed following methods and materials (low $\alpha$ - 1:0.2 + 15% v/v CF + 8% v/v CB and high $\alpha$ - 1:1.8 + 24% v/v tr-x + 8% v/v FS) developed in previous work[5], with the main difference of integration the SF layer (MS 1:1 + 2.8% v/v FS) that encapsulates the MHEC bilayer (see section). The samples each had overall printed width, length, and total thickness of 3 mm, 42 mm, and 1.2 mm, respectively. The samples constitute ~50% v/v of the MHEC bilayer and ~50% of the SF layer. We induce equidistant fractures along the arc length, ranging from 0, 1, 2, 3, 7 fractures per specimen, respectively. Following this, we proceeded to test the electrothermomechanical response of these via Joule heating. The bilayers were electrically connected to a power supply (E36233a, Keysight), where we limit the voltage at 30 V and modulate the current from 150–300 mA with increments of 50 mA, allowing 60 s on each step to reach a steady state temperature. Characterization of the thermal and electromechanical responses is carried out following the same methods presented in previous work[5]. A summary of these studies is shown in Fig. 3.

## Design and testing of stiff, tough, electrically responsive lattices

Lifting robot is constructed using the same materials (low $\alpha$ - 1:0.2 + 15% v/v CF + 8% v/v CB and high $\alpha$ - 1:1.8 + 24% v/v tr-x + 8% v/v FS) and procedures to previous work[5], with the main difference of integrating the SF layer (MS 1:1 + 2.8% v/v FS) to encapsulate the structure (see section). The samples constitute ~70% v/v of MHEC bilayer and ~30% of SF layer. We connected the legs to a power supply (E36233a, Keysight), where we limited the voltage to 30 Volts and modulated the electrical current from 0 mA to 200 mA. Structures were activated at maximum stroke (~6 W) for ~25 s (~1 time constant, see SI Methods), followed by a cut off from power to cool down to room temperature for 120 s ( ~5 time constant, see SI Methods). Characterization of the performance is carried out following procedures described in previous work[5]. The SF layer is consider as external weight and not as an actuator since this does not contribute on the generation of actuation. External weights are normalized by the actuator mass (~65 mg, total mass of 4 legs). A summary of the results from this characterization is presented in Fig. 4.

## Stiff, tough and electrically responsive lattice design, testing, and characterization

Samples are prepared using materials (low $\alpha$ - 1:0.2 + 15% v/v CF + 8% v/v CB and high $\alpha$ - 1:1.8 + 24% v/v tr-x + 8% v/v FS) and procedures detailed in previous work[5], with additional integration of the SF layer (MS

1:1 + 2.8% v/v FS). The structure is printed as a flat square lattice with rib spacing ($\bar{L}$ = 15 mm) and number of cells along the x and y axis ($N_x = N_y = N = 6$), programmed via stereographic projection[3] to transform into a spherical cap with an opening angle of 135°. More details on the lattice design can be found in our previous work on shape shifting structures[3,5]. The samples constitute ~51% v/v of MHEC bilayer and ~49% of the SF layer. We glued thin copper wires (awg 38) with commercially available silver MHEC (MG Chemicals 8331D) to each corner of the lattice, serving as an electrical interface for resistive sensing. A digital multimeter (34460A, Keysight) is used to measure the equivalent electrical resistance of the lattice while testing. After, we mounted it on a uniaxial compression single-axis mechanical tester (Instron 5944 Micro-tester), where compression cycles are induced on the lattice surface. Ten cycles were programmed at a speed of 0.5 mm s$^{-1}$, followed by external impact loads induced with a sledgehammer. A calibrated optical camera (D850, Nikon) was used for imaging the front view of the test sample. This process is repeated six times. A summary of this characterization is shown in Fig. 5b–d. Furthermore, a similar characterization is carried out, where instead of a uniaxial compression test, we drive over the lattice while measuring the equivalent electrical resistance. For this testing, we place the lattice on top of a white paper (for better contrast), and use a calibrated optical camera (D850, Nikon) to image the compression of the lattice. A summary of the results from this testing is presented in Fig. 5e.

## Data availability
A minimum dataset supporting the findings of this study is included in the Supplementary Information. Additional data and code are available from the corresponding author upon request.

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

## Acknowledgments

We gratefully acknowledge support from the National Science Foundation CAREER Award (CMMI-2047683; J.W.B) and the AFOSR Young Investigator Award (FA9550-20-1-0365; J.W.B.), the Department of Defense (DOD) National Defense Science and Engineering Graduate (NDSEG) fellowship program (J.M.M.F and C.K.), and the US Department of Education (GAANN; C.K.). This material is based upon work supported by the National Science Foundation Graduate Research Fellowship under Grant No. 2234657 (N.B.). Any opinion, findings, and conclusions or recommendations expressed in this material are those of the authors(s) and do not necessarily reflect the views of the National Science Foundation. This work used the shared experimental facility BioInterface Technologies (BIT) Facility. We thank the Fluid Lab, especially Professor J. Bird and G. Lee, for technical assistance on thermal imaging. We thank R. Sanchez and X. Ye for their helpful discussion and technical assistance.

## Author contributions

J.W.B. and J.M.M.F. conceived the study. C.K. performed the chemical characterization of the adhesive materials. N.B. conducted experiments and data capture for imaging of the printing routines for all specimens. J.M.M.F. was responsible for the remaining design, studies, and analyses. J.W.B. supervised the project.

## Competing interests

The authors declare no competing interests.

## Additional information

**Peer review information** : *Nature Communications* thanks the anonymous reviewers for their contribution to the peer review of this work. A peer review file is available.

