## [Transparent Peer Review file · Nature Communications]

Synthetic fascia for stiff and tough 4D printed multifunctional structures that detect and tolerate damage

Corresponding Author: Professor John Boley

Version 0:

Reviewer comments:

Reviewer #1

(Remarks to the Author)

The authors present a continuation of their previous work on multiscale heterogeneous epoxy composites (MHECs) by introducing a co-printed PDMS-based "synthetic fascia" to enhance toughness and damage tolerance in 4D printed structures. The manuscript is well-organized and the experimental work is carefully executed, demonstrating improved mechanical resilience, self-sensing, and partial retention of functionality after mechanical failure.

However, the contribution appears incremental. The core materials platform, fabrication method, and actuation mechanism (Joule heating) are largely unchanged from the prior study. The central novelty—the addition of a soft, co-printed fascia layer—is effective and biomimetic but follows well-established strategies in soft material toughening. While the added damage tolerance is useful, it represents a performance optimization rather than a conceptual or methodological advance. As such, I do not believe the manuscript meets the novelty threshold required for publication in Nature Communications, even if framed as a platform extension.

Comments for the authors:

- The manuscript repeatedly refers to "constant power" actuation using a Keysight E36233A power supply. However, this instrument does not support constant power mode; it operates in either constant voltage or constant current. Based on the methods described, the authors likely performed constant current actuation by manually stepping current at a fixed voltage. This is a critical distinction: in thermally responsive systems with dynamic resistance, true constant power is required to avoid uncontrolled self-heating and thermal runaway. A brief survey of the literature (e.g., 10.1002/admt.202301769) shows that others have used dedicated power-regulated systems to address this challenge. I strongly recommend the authors either (1) revise their terminology throughout the manuscript to reflect constant current operation, or (2) repeat the relevant experiments using a power source that supports true constant power output. The current description is technically inaccurate and may misrepresent the thermal behavior of the actuators.

- The manuscript claims record-setting lifting performance and actuation stress (e.g., lifting 1230× the actuator's own weight). However, there is no benchmarking table or standardized comparison with other reported actuators or artificial muscles. Quantitative comparisons of energy density, power density, and cycle life against state-of-the-art actuators are essential to support these claims.

- The authors should explicitly assess long-term actuation performance under realistic conditions. I recommend conducting at least 1000 continuous actuation cycles using true constant power input, with tracking of curvature, response time, and degradation over time. Many Joule-heated polymer actuators degrade significantly after tens to hundreds of cycles, and without this data, the practical utility of the proposed system remains uncertain.

- Finally, the terminology around "damage tolerance" vs. "self-healing" should be used with precision. The system does not demonstrate self-healing in the material science sense (e.g., polymer network reformation). Rather, the "fascia" structure enables structural containment and temporary mechanical function post-fracture. Clarifying this distinction will help avoid overstating the material's capabilities.

Overall, the work is technically sound and addresses an important limitation of the previous study. However, to warrant

publication in a high-impact journal, the manuscript requires greater differentiation, clearer terminology, and more rigorous performance evaluation.

Reviewer #2

(Remarks to the Author)

This increased toughness of this composite of composites is very interesting and an excellent application of multi-material printing capabilities of DIW. The material engineering presented from toughening of the matrix materials to the adhesion of the composite is elegant and well described.

The importance of the work is sufficient for publication in Nat. Comm., as it is a very tough, switchable composite and the biomimetic comparisons are intriguing.

The description of the work, however, is confusing. If I understand correctly, there is not active control of motion; rather, there is active control of stiffness. Also, the term "4D" I find is hyperbolic and unnecessary. Particularly if I am correct in understanding that the motion comes from equilibrating with the environment rather than using energy to go out of equilibrium. I think the science and engineering would stand out more with better clarity in the contribution, and reduction in hype words.

If I am incorrect and these actuators cause motion, please make that more clear in the writing – and distinguish it as an out of equilibrium actuation mode.

This clarity could come quite easily:

(i) The performance of prior MHECs are described in the introduction, but the mechanism is not clearly described.

Referencing the publication is sometimes OK for efficiency, but I think it is important enough to briefly describe in the main text.

(ii) Page 6, line 156 – please explain the mechanism a little bit in the text instead of just a reference.

(iii) Page 3, line 85 – what does the pronoun "this" refer to?

With these changes, I would be delighted to re-review.

Thank you

Reviewer #3

(Remarks to the Author)

Boley and coworkers reported a strategy to print PDMS-epoxy composite structures. These structures exhibit high stiffness and high toughness at the same time. The authors characterized the mechanical properties of the composite through tensile test and measured the adhesion strength between the two component materials through peeling test. The authors printed a bilayer structure that can bend when a voltage is applied and designed an actuator that can lift weight. The second demonstration was a lattice structure that can be used as an impact sensor. The manuscript is technically sound and well-organized. The proposed strategy also showed novelty to some extent. The reviewer recommends the publication of this manuscript if the following concerns can be properly addressed:

Fig 1c error bar color is too faint.

Fig 3b, the reviewer is confused about how to understand the pattern shown in the plot. The authors should properly clarify what does the number of peaks mean and how to interpret the magnitude of the peaks.

Fig 4e, how is the actuation stress calculated? The calculation of actuation stress should depend on the type of the actuator, e.g. contraction-type or bending type. Actuation stress needs a clearer definition before comparisons can be made.

Fig 4c,d the black points seem redundant.

Fig 5c, the authors claimed to make a force sensor. How is the R/R0 correlated with the compressive force? In the first cycle, the R/R0 change is very small, after fifth impact, the R/R0 change becomes much larger, though the compressive force becomes even smaller. Is this feature desired for a force sensor?

Fig 5d is not clear as all data points overlap.

Version 1:

Reviewer comments:

Reviewer #1

(Remarks to the Author)

The authors have addressed my earlier concerns regarding technical execution and clarification of experimental methods. However, I remain unconvinced about the novelty of the contribution. While the integration of the fascia layer improves performance, the core concepts and mechanisms remain largely consistent with the authors' previous work. As such, the advance feels incremental rather than foundational.

Reviewer #3

(Remarks to the Author)

Authors have addressed my concerns. Publication is suggested.

Dear Editors and Reviewers,

Below is a point-by-point list of how we addressed each reviewer comment in our revised submission. We want to thank the reviewers for their important feedback, which we believe has significantly improved the impact and quality of our work.

Sincerely, on behalf of the authors

-Will Boley

Responses to the Reviewers

A guide to our color convention:

- Original feedback from the Editor and Reviewers are in **black**,
- Responses to the Editor and Reviewers are in **blue**,
- New/modified text that we incorporated into the manuscript and Supplementary Information (SI) in response to the Editor and Reviewers are italicized and in *teal*,
- Additional text incorporated into the manuscript and SI are italicized and in *red*.

Responses to Reviewer #1

The authors present a continuation of their previous work on multiscale heterogeneous epoxy composites (MHECs) by introducing a co-printed PDMS-based “synthetic fascia” to enhance toughness and damage tolerance in 4D printed structures. The manuscript is well-organized and the experimental work is carefully executed, demonstrating improved mechanical resilience, self-sensing, and partial retention of functionality after mechanical failure.

However, the contribution appears incremental. The core materials platform, fabrication method, and actuation mechanism (Joule heating) are largely unchanged from the prior study. The central novelty—the addition of a soft, co-printed fascia layer—is effective and biomimetic but follows well-established strategies in soft material toughening. While the added damage tolerance is useful, it represents a performance optimization rather than a conceptual or methodological advance. As such, I do not believe the manuscript meets the novelty threshold required for publication in Nature Communications, even if framed as a platform extension.

We thank the reviewer for their thoughtful and detailed evaluation. We appreciate the recognition of the quality and careful execution of the experimental work, as well as the acknowledgment of the improved mechanical resilience, self-sensing, and functional retention demonstrated by the system.

While we agree that the current study builds upon our previously established MHEC platform, we respectfully emphasize that the co-printed synthetic fascia introduces a meaningful advance in the field of damage-tolerant actuators. Specifically, the novelty lies not solely in the addition of a soft layer, but in its strategic co-design and integration with the underlying heterogeneous actuator, enabling simultaneous improvements in mechanical toughness, damage tolerance, and post-fracture functionality — all without compromising actuation stiffness, performance, or sensing capability. A key distinction of this work is that such toughness enhancement via soft adhesive interlayers is typically demonstrated in non-functional or passive materials, whereas our system achieves this while preserving full actuation and self-sensing functionality. Furthermore, the approach is generalizable and can be extended to a wide range of responsive actuator materials. This positions the work as a platform-level enhancement rather than a simple material substitution or marginal optimization.

1. The manuscript repeatedly refers to “constant power” actuation using a Keysight E36233A power supply. However, this instrument does not support constant power mode; it operates in either constant voltage or constant current. Based on the methods described, the authors likely performed constant current actuation by manually stepping current at a fixed voltage. This is a critical distinction: in thermally responsive systems with dynamic resistance, true constant power is required to avoid uncontrolled

self-heating and thermal runaway. A brief survey of the literature (e.g., 10.1002/admt.202301769) shows that others have used dedicated power-regulated systems to address this challenge. I strongly recommend the authors either (1) revise their terminology throughout the manuscript to reflect constant current operation, or (2) repeat the relevant experiments using a power source that supports true constant power output. The current description is technically inaccurate and may misrepresent the thermal behavior of the actuators.

We thank the reviewer for this important and technically insightful comment. We acknowledge that our use of the term "constant power" to describe our actuation conditions was imprecise and potentially misleading. To clarify, we operated the system using the Keysight E36233A power supply in constant voltage mode, setting the voltage to 30 V and limiting the maximum current to a set point target based on the specific demonstration (i.e., bilayers, lifting robot), thereby ensuring that the power output remained approximately constant during operation. We did not use a true constant power control mode, and we agree that this distinction is critical, particularly for thermally responsive systems with dynamic resistance characteristics. We have thoroughly revised the manuscript to correct the terminology.

Specifically, in the "Damage-tolerant heavy lifting robot" section, we corrected the statement:

Original: "The actuation cycle commenced with a constant power input (6 W) sustained for approximately 25 seconds..."

Revised: "*The actuation cycle commenced with a fixed voltage input of 30 V, with the current limited to a maximum of 0.2 A, resulting in an approximate power output of 6 W sustained for approximately 25 seconds...*"

We also include the following statement in the Methods section, specifically within the Characterizing Damage-Tolerant Electromechanical Bilayers subsection, to clarify the voltage and current parameters used during bilayer testing.

The bilayers were electrically connected to a power supply (E36233a, Keysight), where we limit the voltage at 30 V and modulate the current from 150 – 300 mA with increments of 50 mA, allowing 60 seconds on each step to reach a steady state temperature.

We believe these revisions improve the technical accuracy and clarity of the manuscript, and we appreciate the reviewer bringing this to our attention.

2. The authors should explicitly assess long-term actuation performance under realistic conditions. I recommend conducting at least 1000 continuous actuation cycles using true constant power input, with tracking of curvature, response time, and degradation over time. Many Joule-heated polymer actuators degrade significantly after tens to hundreds of cycles, and without this data, the practical utility of the proposed system remains uncertain.

We thank the reviewer for this important suggestion regarding the assessment of long-term actuation performance, which is indeed a critical consideration for the practical deployment of Joule-heated actuators. We would like to note that in our previous work, we conducted extensive cyclic testing of the underlying multiscale heterogeneous epoxy composite (MHEC) system without the integration of the synthetic fascia.

In that study, we demonstrated stable actuation performance over more than 20,000 continuous cycles, highlighting the repeatable and reliable actuation and self-sensing mechanisms of these actuators, thereby confirming the durability of the core material platform under Joule heating.

In the current study, our focus was on the introduction of the co-printed PDMS-based synthetic fascia, designed to enhance toughness, damage detection and tolerance, and functional retention under extreme mechanical deformation. While the synthetic fascia significantly improves mechanical robustness, it is mechanically integrated yet electrically isolated from the Joule-heated regions. Therefore, it does not directly influence the Joule heating response, thermal actuation mechanism, or actuation repeatability. Based on the actuator's performance, we do not expect the cycling life to be negatively affected by the inclusion of the fascia.

As shown in Figure 3e and 3f, even with the addition of the fascia, the actuator maintains the expected actuation response. This is consistent with the fact that the Young’s modulus of the synthetic fascia is significantly lower than that of the epoxy actuator, and therefore, it does not constrain the actuation.

Furthermore, the damage-tolerant characteristics provided by the fascia are expected to mitigate the effects of thermal fatigue and mechanical degradation, potentially improving long-term operational reliability.

We have added the following statement on section *Self-Sensing, Damage-Tolerant Bilayer Actuators* for clarity:

Moreover, based on our previous work demonstrating stable performance over more than 20,000 actuation cycles for the underlying MHEC system, and considering the electrically and mechanically decoupled nature of the synthetic fascia, we expect comparable or improved long-term durability for the integrated structure.

3. Finally, the terminology around "damage tolerance" vs. "self-healing" should be used with precision. The system does not demonstrate self-healing in the material science sense (e.g., polymer network reformation). Rather, the "fascia" structure enables structural containment and temporary mechanical function post-fracture. Clarifying this distinction will help avoid overstating the material’s capabilities.

We thank the reviewer for this important comment regarding the terminology around "damage tolerance" and "self-healing." We have carefully reviewed the manuscript and confirm that we do not use the term "self-healing" to describe our system. Instead, we consistently refer to the role of the synthetic fascia as providing damage tolerance, emphasizing structural containment and the retention of partial mechanical functionality post-fracture without implying polymer network reformation or intrinsic material self-repair.

To ensure clarity and avoid any potential misunderstanding, we have double-checked the manuscript and ensure that the term self-healing is not use in any of the sections.

Overall, the work is technically sound and addresses an important limitation of the previous study. However, to warrant publication in a high-impact journal, the manuscript requires greater differentiation, clearer terminology, and more rigorous performance evaluation.

We sincerely thank the reviewer for their constructive feedback and thoughtful assessment of our work. We appreciate the recognition of the technical soundness of the study and its relevance in addressing an important limitation of our previous work. In response to the reviewer’s suggestions, we have carefully revised the manuscript to improve clarity, strengthen the differentiation from our prior work, and ensure precise and consistent terminology throughout. We have also clarified the scope and practical implications of the performance evaluation presented. We believe these revisions have significantly improved the quality and clarity of the manuscript, and we thank the reviewer for their valuable input, which has helped us enhance the presentation and impact of this work.

Responses to Reviewer #2

This increased toughness of this composite of composites is very interesting and an excellent application of multi-material printing capabilities of DIW. The material engineering presented from toughening of the matrix materials to the adhesion of the composite is elegant and well described. The importance of the work is sufficient for publication in Nat. Comm., as it is a very tough, switchable composite and the biomimetic comparisons are intriguing.

We thank the reviewer for their supportive and thoughtful comments. We appreciate the recognition of the composite’s enhanced toughness, the effective use of multi-material DIW, and the elegance of the material integration strategy. We are pleased that the biomimetic comparisons and the multifunctional performance were found compelling and that the significance of the work aligns with the scope of Nature Communications.

1. The description of the work, however, is confusing. If I understand correctly, there is not active control of motion; rather, there is active control of stiffness. Also, the term “4D” I find is hyperbolic and unnecessary. Particularly if I am correct in understanding that the motion comes from equilibrating with the environment rather than using energy to go out of equilibrium. I think the science and engineering would stand out more with better clarity in the contribution, and reduction in hype words. If I am incorrect and these actuators cause motion, please make that more clear in the writing – and distinguish it as an out of equilibrium actuation mode.

We thank the reviewer for the opportunity to clarify these important points.

To clarify, our actuators do indeed exhibit active motion, rather than solely stiffness modulation. The motion is driven by Joule heating, which induces a controlled temperature increase within the composite. This electrically activated thermal input causes bending due to anisotropic expansion or contraction due to the mismatch in coefficients of thermal expansion (CTEs) across the bilayer structure that composes the actuators. Based on Timoshenko’s theory for bimetallic thermostats, we can accurately model and predict the resulting curvature as a function of temperature and time.

Regarding the term “4D printing,” we respectfully note that its use is well-established in the active materials and additive manufacturing community. It describes 3D printed structures designed to undergo time-dependent transformations in response to external stimuli such as temperature, light, humidity, or magnetic fields. Our system fits this definition, as the printed composite exhibits reversible, stimulus-responsive shape change upon electrical activation.

To further improve clarity, we have included the following statements in the revised manuscript, specifically in the introduction section:

Actuation mechanism: The mechanism of these actuators is governed by Timoshenko’s theory for bimetallic thermostats, wherein a bilayer composed of two stiff materials with differing coefficients of thermal expansion (CTEs) undergoes curvature change due to thermal mismatch. This CTE mismatch induces bending upon Joule heating, resulting in a predictable and programmable shape transformation. Depending on the application, individual bilayers can operate as standalone actuators or be arranged into systematic lattice configurations to enable more complex, multi-curvature shape changes.

4D printing: 4D printing refers to an advanced form of additive manufacturing in which a 3D printed object is designed to undergo controlled, time-dependent changes in shape, properties, or function in response to external stimuli such as temperature, light, humidity, magnetic fields, or other environmental factors. This stimulus-responsive behavior distinguishes 4D printing from conventional static 3D structures, enabling dynamic and functional transformations. While many 4D printed systems are designed for reversible shape changes, certain applications exploit irreversible transformations to achieve stable, deployed configurations, expanding the functional scope of 4D printed architectures.

2. The performance of prior MHECs are described in the introduction, but the mechanism is not clearly described. Referencing the publication is sometimes OK for efficiency, but I think it is important enough to briefly describe in the main text.

We thank the reviewer for this important suggestion. To improve clarity and provide a more self-contained description, we have expanded the introduction to briefly summarize the actuation mechanism of the prior MHECs. Specifically, we now include the following explanation: The mechanism of these actuators is governed by Timoshenko’s theory for bimetallic thermostats, wherein a bilayer composed of two stiff materials with differing coefficients of thermal expansion (CTEs) undergoes curvature change due to thermal mismatch. This CTE mismatch induces bending upon Joule heating, resulting in a predictable and programmable shape transformation. Depending on the application, individual bilayers can operate as standalone actuators or be arranged into systematic lattice configurations to enable more complex, multi-curvature shape changes.

This addition ensures the manuscript is more accessible and clear to readers unfamiliar with our prior work, while maintaining appropriate referencing.

3. Page 6, line 156 – please explain the mechanism a little bit in the text instead of just a reference.

We thank the reviewer for highlighting the importance of explaining the actuation mechanism in the main text rather than relying solely on a citation. In response, we have revised the manuscript (Page 6, Line 156) to include the following statement:

For the core MHEC bilayer, we selected a combination of two stiff epoxy-based inks with distinct CTEs. The mismatch in CTEs enables curvature changes upon heating, as described by Timoshenko’s theory for bimetallic thermostats. To enable active control via Joule heating, at least one of the inks must be electrically conductive. This allows us to locally and precisely regulate temperature through electrical input. For additional details on the design and composition of the electrically responsive MHEC bilayer, we refer the reader to our previous work.

This addition provides a clear, concise explanation of the actuation mechanism while still directing readers to our prior publication for more in-depth technical details.

We sincerely thank the reviewer for their constructive and thoughtful feedback. We have carefully addressed all comments, clarifying terminology, improving descriptions of the actuation mechanism, and refining our presentation of the novelty and technical significance of the work. These suggestions have helped us significantly strengthen the manuscript, and we appreciate the opportunity to improve the clarity, rigor, and impact of our study.

Responses to Reviewer #3

Boley and coworkers reported a strategy to print PDMS-epoxy composite structures. These structures exhibit high stiffness and high toughness at the same time. The authors characterized the mechanical properties of the composite through tensile test and measured the adhesion strength between the two component materials through peeling test. The authors printed a bilayer structure that can bend when a voltage is applied and designed an actuator that can lift weight. The second demonstration was a lattice structure that can be used as an impact sensor. The manuscript is technically sound and well-organized. The proposed strategy also showed novelty to some extent. The reviewer recommends the publication of this manuscript if the following concerns can be properly addressed:

We sincerely thank the reviewer for their positive evaluation and thoughtful comments on our work. We appreciate the recognition of the novelty and technical merit of our approach. In response to the reviewer’s concerns, we have provided detailed point-by-point responses and made corresponding revisions in the manuscript to address each comment.

1. Fig 3b, the reviewer is confused about how to understand the pattern shown in the plot. The authors should properly clarify what does the number of peaks mean and how to interpret the magnitude of the peaks.

We thank the reviewer for pointing out the need for clarification. In Figure 3b, each isolated set of peaks corresponds to a fracture event induced manually on a single bilayer sample. At lower peaks (from $N = 1$ and $N = 3$), we observe a single peak for each new fracture, with a magnitude increasing with N . From $N = 4$ to $N = 5$ we see a single high magnitude primary peak, likely resulting from the new fracture, and the presence of additional secondary peaks, which is likely a result of small motions at the interfaces of the existing fractures. Above $N = 5$ we see multiple peaks of similar magnitude, which is likely a result of the additional fracture and collective interfacial slipping across the multiple existing fractures of the sample. We have revised the corresponding section in the manuscript to explicitly state this interpretation for improved clarity. This is the statement that we added:

These fractures are manually induced on a single bilayer sample in a sequential manner. Each set of peaks in the signal corresponds to a new fracture event. At lower peaks (from $N = 1$ and $N = 3$), we observe a single peak for each new fracture, with a magnitude increasing with N . From $N = 4$ to $N = 5$ we see a single high magnitude primary peak, likely resulting from the new fracture, and the presence of additional secondary peaks, which is likely a result of small motions at the interfaces of the existing fractures. Above $N = 5$ we see multiple peaks of similar magnitude, which is likely a result of

the additional fracture and collective interfacial slipping across the multiple existing fractures of the sample.

2. Fig 4e, how is the actuation stress calculated? The calculation of actuation stress should depend on the type of the actuator, e.g. contraction-type or bending type. Actuation stress needs a clearer definition before comparisons can be made.

We thank the reviewer for raising this important point regarding the definition of actuation stress. In our study, actuation stress is defined as the lifted mass divided by the cross-sectional area of the actuator. We have clarified this definition in the revised manuscript to ensure transparency and consistency when comparing actuation performance. This is the statement that was added to the manuscript:

Actuation stress is defined as the lifted weight divided by the cross-sectional area of the actuator.

3. Fig 4c,d the black points seem redundant.

We thank the reviewer for the observation. The black points in Figures 4c and 4d represent failure points and are intentionally included to highlight the resilience and damage tolerance of our approach. Showing these points emphasizes the actuator's performance even after mechanical failure events, which is a key aspect of our study. We have included the following statement in the manuscript to explicitly highlight this:

In **Figure 4c-d**, the measured performance of these actuators is shown with respect to lifted mass for different testings, where blue, yellow, and black dots represent the lifting robot before, after, and right at the mechanical failure, respectively.

4. Fig 5c, the authors claimed to make a force sensor. How is the R/R0 correlated with the compressive force? In the first cycle, the R/R0 change is very small, after fifth impact, the R/R0 change becomes much larger, though the compressive force becomes even smaller. Is this feature desired for a force sensor?

We thank the reviewer for this insightful question. The normalized resistance change (R/R0) in our 3D sensing surface correlates with the compressive force through deformation-induced changes in the conductive network within the actuator. Initially, small resistance changes correspond to minor structural damages or fractures under impact. As the number of impact cycles increases, microstructural damage accumulates, leading to larger resistance changes even under reduced compressive force due to changes in the contact area between the conductive ribs composing the lattice.

This behavior reflects a form of damage sensing rather than a purely linear force-resistance relationship. While the sensor's sensitivity increases with repeated impacts due to damage accumulation, this feature is beneficial for applications requiring detection of cumulative mechanical stress or fatigue rather than instantaneous force measurement alone.

Importantly, the main goal of this structure is to increase toughness and resilience. By incorporating an electrically conductive core, we also enable multifunctionality: a structure with high stiffness, mechanical resilience, and the capability to sense damage. This technology could potentially be used for surface damage mapping or cumulative damage detection in real-time.

We refer the reviewer to Section 3D Sensing Surface That Detects and Tolerates Damage, where these statements are found:

Notably, it is observed that the force required to compress the lattice decreases with each subsequent external perturbation (e.g., hits with a sledgehammer), indicating the introduction of fractures in the lattice structure. This observation aligns with the spikes in the change in electrical resistance (R/R0), coinciding with the application of external perturbations, showcasing an inherent fracture detection capability. An intriguing behavior is evident in the electrical responses of the lattice: as additional fractures are induced, the sensitivity of the resistance in response to deformation increases. This phenomenon is attributed to the amplified change in resistance resulting from slight relative movements between segments of the fractured lattice ribs. These movements reduce the contact area within the fractured ribs, thus increasing the electrical resistance with respect to an external force. However,

since the synthetic fascia surrounding the electrically conductive ribs is hyperelastic and adheres to the MHEC ribs, after the compressive loads are released, the MHEC ribs can revert to their initial configuration, increasing the contact area of the ribs and thereby reducing the electrical resistance of the lattice close to its nominal value. Prior research on soft sensors composed of conductive fillers provides additional support for these findings [52].

5. Fig 5d is not clear as all data points overlap.

We thank the reviewer for this observation. The overlapping data in Fig. 5d is intentional and serves to highlight the evolution of electrical response as damage accumulates in the 3D surface. Specifically, this presentation allows for a direct visual comparison of performance across different damage states. By overlaying the curves, we aim to demonstrate how the 3D surface can exhibit different sensing responses depending on the degree of induced damage — a feature that can be tailored based on the application.

To further clarify this, we have edited part of the relevant section titled 3D Sensing Surface That Detects and Tolerates Damage, where we state:

A summary of these findings is presented in Figure 5d. The synthetic muscle composite lattice exhibits varied force and resistance responses, depending on the number of induced fractures (i.e., the number of external perturbations from the sledgehammer). At a low number of external perturbations (e.g., three or less), the surface exhibits a relatively high compressive force and low electrical responsiveness. At a higher number of external perturbations (e.g., four or more), the surface exhibits a low compressive force and high electrical response. Furthermore, this synthetic muscle composite demonstrates high resilience to various external perturbations, showcasing 3D surfaces capable of sensing and damage detection.

We sincerely thank the reviewer for their thoughtful and constructive feedback. We greatly appreciate the time and effort taken to evaluate our work and provide detailed comments. In response, we have carefully addressed each point raised and revised the manuscript accordingly to improve clarity, technical accuracy, and overall presentation. Specific responses to each comment, along with the corresponding changes in the manuscript, are provided below. We believe these revisions have significantly strengthened the manuscript and we hope it now meets the expectations for publication.